# Relationship between Blood Parameters and Outcome in Rescued Roe Deer

**DOI:** 10.3390/ani12243469

**Published:** 2022-12-08

**Authors:** Maria Irene Pacini, Francesca Bonelli, George Lubas, Micaela Sgorbini

**Affiliations:** Department of Veterinary Sciences, University of Pisa, Viale delle Piagge 2, 56100 Pisa, Italy

**Keywords:** haematology, biochemistry, roe deer, wildlife, blood analysis

## Abstract

**Simple Summary:**

Given that wild animal hospitalisations are increasing, veterinary knowledge of wildlife medicine needs to be improved along with more effective clinical and diagnostic procedures. A blood test can be a quick and minimally invasive way of gathering useful clinical information. This study evaluates the haematological and biochemical parameters in injured roe deer and discusses their values in relation to animal hospitalisation outcomes. The study was conducted on a cohort of 98 roe deer divided into groups based on age and hospitalisation outcome. A panel of haematological and biochemical parameters was performed for each animal. Significant differences were found within both the unweaned and weaned groups. Creatine kinase, creatinine, and bilirubin may be useful indicators to correlate with the severity of trauma.

**Abstract:**

Veterinary facility admissions of wild animals are increasing alongside the interest in wildlife diseases. To improve animal welfare, it is therefore important to increase veterinarians’ knowledge of wild animal medicine and to improve the clinical and diagnostic procedures, especially in the case of patients affected by trauma or multiple traumas. Blood analysis can be a quick and minimally invasive way of gathering useful clinical information for adequate treatment and management, and, together with a good clinical examination, to help predict hospitalisation outcomes. Few papers have reported reference ranges for the haematological and biochemical parameters of roe deer. This study evaluates the haematological and biochemical parameters in traumatised roe deer in relation to animal hospitalisation outcomes. The study was carried out on a cohort of 98 roe deer divided into groups according to their age and hospitalisation outcome. For each animal, a panel of haematological and biochemical parameters was performed. Significant differences were found between unweaned (<4 months old) groups in terms of MCV, MCH, CK, creatinine, total bilirubin, direct bilirubin, and indirect bilirubin, and between weaned (>4 months old) groups for total bilirubin. Creatine kinase, creatinine, and bilirubin may be useful indicators to correlate with the severity of trauma and help predict prognosis.

## 1. Introduction

Roe deer are wild ruminants belonging to the Cervidae family. They are widely distributed in Europe, with an estimated population of more than 426,000 roe deer in Italy, mostly in the Alps and central-northern Apennines [1].

Tuscany (central Italy) has 400,000 wild ungulates, more than any other region, consisting above all of roe deer, red deer, fallow deer, wild boar, and mouflon. The population of roe deer in Tuscany is estimated to be about 130,000 animals [1,2].

In recent decades, there has been a demographic increase in wild ungulates, including roe deer. This is due to the changes in wildlife management, such as the increase in the regulation of hunting and protected areas, and the centralisation of economic activities in cities with the resulting abandonment of the countryside and landscape change. Their behavioural and nutritional plasticity have enabled roe deer to colonise abandoned territories, thus rapidly increasing their home range from the native forest habitat to the countryside and even suburban areas [3,4,5].

The growth in the wildlife population, urban planning, and landscape fragmentation have resulted in increased interactions between wildlife, humans, and domestic animals. This has led to increasing injuries and even deaths both for humans and wild ungulates, and the number of wild ungulates admitted to wildlife rescue centres and veterinary facilities has increased. The most common reason for hospitalisation is collisions with vehicles, but also becoming trapped in metal fences or nets, predation by dogs, the unjustified rescue of fawns, and trauma from agricultural machines also occur [6,7,8].

The road network is a barrier to the movement of wild animals and a threat to their survival. Roads break up the home range of wild species and hinder their movements and dispersive migrations. The animals therefore need to cross the roads in search of food, territories, and partners for mating, thus endangering their own lives and also those of the drivers [9,10]. The number of road accidents involving wildlife is constantly increasing in all industrialised countries, with several million cases per year in Europe [9,11,12] and several thousand in Italy [10,13]. Recent data on road accidents involving wild ungulates in Europe clearly show a dramatic increase and between 750,000 and one million are estimated to occur annually [14,15].

In Italy, most wildlife accidents concern wild boar and roe deer, followed by badgers, foxes, red deer, fallow deer, hares, and porcupines [13,16]. In Tuscany, 188 road accidents were recorded in 2001 and 478 in 2008 [13,17]; however, more recent data are not available. Usually, the wounded roe deer rescued and admitted to a veterinary facility are severely injured and show multiple traumas. Wild ungulates are also very susceptible to developing capture myopathy, a severe myositis typical of wild ruminants and consisting of muscle damage induced by stressful manipulations [18], such as capture, rescue, transport, and restraint [19,20]. For these reasons, the mortality rate of injured roe deer is very high [6,8,21,22].

Considering the close interaction between humans and wildlife populations, the admission of wild animals to veterinary facilities is continuing to increase alongside the interest in wildlife diseases [23,24,25,26,27]. It is therefore important to increase veterinarians’ knowledge of wild animal medicine and to improve clinical and diagnostic procedures. In the case of animals affected by trauma or multiple traumas, it is essential to act promptly upon admission to assess their clinical condition.

Among the diagnostic tests, blood analysis is a rapid and minimally invasive procedure that requires short-term contact with the animal and minimal stress. In conjunction with examination, haematological and biochemical blood parameters can provide additional clinical information, which can be used to determine case-specific treatment, management, and prognosis following hospitalisation.

There are few papers regarding the reference ranges for the haematological and biochemical parameters of roe deer [19,28,29,30,31]. The aim of this study was thus to evaluate the haematological and biochemical parameters in traumatised roe and to discuss their values in relation to the animals’ hospitalisation outcomes.

## 2. Materials and Methods

### 2.1. Study Population and Sample Collection

The study was carried out on a cohort of 98 roe deer rescued in the Pisa district and referred to our Veterinary Teaching Hospital (VTH) between 2010 and 2021.

The animals were divided into groups according to their age and hospitalisation outcome. Two different age classes were identified: unweaned (<4 months old) and weaned (>4 months old). The age was based on morphological aspects and on a dental examination to determine the state of eruption and wearing down of the teeth [32]. The animals were retrospectively subtyped based on their outcome as deceased (defined as spontaneous death during the hospitalisation period or euthanised because they were near death and to reduce pain and suffering or because they were not compatible with a positive release in the wild) or survived (defined as fully recovered from the injury and released back into their wildlife environment or given in custody to authorised people). According to the parameters described above, all roe deer were divided into the following groups: deceased unweaned (Group A1); survived unweaned (Group A2); deceased weaned (Group B1); survived weaned (Group B2).

At admission to our VTH, each animal was placed in an individual box with a straw litter in a closed building accessible only to authorised personnel. The animal’s history was collected from the person who transported the animal from the rescue area to the VTH. Each animal underwent a brief clinical examination that lasted only a few minutes, and the age was assessed from the dental status. Since the animals were difficult to handle or required diagnostic imaging procedures or surgery, the clinical examination, blood collection, and all other medical manipulations were performed under sedation or general anaesthesia as previously reported [7].

### 2.2. Blood Analysis

Blood samples were collected from the jugular, cephalic, or saphenous veins, as soon as possible after the clinical examination and always before any therapy, in K3-EDTA (ethylenediaminetetraacetic acid) and heparinised (LH) tubes. The use of lithium–heparin as anticoagulant was to limit the probability of haemolysis that might otherwise have interfered with the analysis.

Specimens containing clots, partially or highly haemolysed (according to the Clinical and Laboratory Standards Institute, using visual sample inspection and haemoglobin estimated over 150 mg/dL) [33] were excluded from the analysis.

The K3-EDTA samples were processed within 15 min of collection for a complete blood count (CBC) using an automated laser cell counter (Procyte, Idexx Laboratories, Westbrook, ME, USA) set to “other species” and the following parameters were assessed: red blood cell count (RBC), white blood cell count (WBC), haemoglobin (Hgb), haematocrit (Hct), mean corpuscular volume (MCV), mean corpuscular haemoglobin (MCH), mean corpuscular haemoglobin concentration (MCHC), and platelet count (PLT). The K3-EDTA was also used for blood smear preparation. The slides were air-dried and stained with the May–Grundwald Giemsa stain using an automatic slide stainer (Aerospray Hematology Slide Stainer mod. 7150, Delcon, Italy). The staining time was 12 min. The differential cell counts were performed by a board-certified clinical pathologist (G.L.).

The heparinised samples were centrifuged at 2100 rpm for 10 min and the harvested heparinised plasma was used to assess the following parameters using a spectrophotometric and immunoturbidimetric analyser (Liasys, Analyzer Medical System-AMS, Aprilia, Latina, Italy) and dedicated reagents kits: total protein (PT) (biuret colorimetric method), creatinine (kinetic modified Jaffè method), urea (kinetic enzymatic method), and total and direct bilirubin (colorimetric method without DMSO concentrations). The indirect bilirubin value was calculated from the value of total bilirubin minus the value of direct bilirubin. Aspartate aminotransferase (AST) (kinetic method UV—IFCC), gamma glutamyl-transferase (GGT) (kinetic method-Szasz-Tris), and creatine kinase (CK) (kinetic method UV) activities were also assessed.

### 2.3. Statistical Analysis

The data distribution was evaluated using the Kolmogorov–Smirnov test for normality, and the results were expressed as median, and minimum and maximum values. The Mann–Whitney U test for unpaired data was applied to assess the statistical differences for each group between animals that had died or survived. The Spearman rank correlation test was applied between MCH and vs. creatinine, total bilirubin, and creatine kinase. Statistical significance was set at *p* < 0.05.

## 3. Results

A total of 98 roe deer, 16/98 (16.3%) unweaned and 82/98 (83.7%) weaned, were grouped according to their age and hospitalisation outcome into group A1 (deceased unweaned), group A2 (survived unweaned), group B1 (deceased weaned), and group B2 (survived weaned), as reported in Table 1. The reasons for hospitalisation are reported in Table 2.

Overall, 70/98 (71%) patients were rescued due to collision with a vehicle.

The clinical lesions diagnosed in the cohort of 98 roe dears are reported in Table 3.

Overall, 66/77 (86%) deceased animals (Groups A1 and B1) were diagnosed with fracture/luxation of limbs, pelvis, vertebrae, or multiple traumas, while most of the survived patients were affected by soft tissue injuries or had no lesions at all (14/21, 67%).

Haematological analyses were performed in 86/98 (88%) animals: 8/16 (50%) in Group A1, 6/7 (86%) in Group A2, 59/68 (87%) in Group B1, and 13/14 (93%) in Group B2. Haematological analyses were not performed in 12/98 (12.2%) animals.

Biochemical parameters were assessed in all of the animals enrolled. The haematological and biochemical parameter values are reported in Table 4 and Table 5, respectively.

Statistical differences revealed by the Mann–Whitney test were found between Group A1 and A2 for MCV (*p* = 0.0350), MCH (*p* = 0.0029), CK (*p* = 0.0006), creatinine (*p* = 0.0182), total bilirubin (*p* = 0.0025), direct bilirubin (*p* = 0.0120), and indirect bilirubin (*p* = 0.0123). A significant difference was found between Group B1 and B2 for total bilirubin (*p* = 0.0206). With the Spearman test, a correlation was found between MCH and total bilirubin in Group A1 (r = 0.8857; *p* = 0.0333) and Group A2 (r = 0.0159; r = −0.8729).

## 4. Discussion

This paper reports the data from haematological and biochemical analyses performed over an 11-year period on traumatised roe deer rescued in the area around Pisa (central Italy).

Overall, the survival rate in the unweaned groups was higher than in the weaned groups, mainly due to the different causes of hospitalisation. Most of the animals in Group B were hospitalised following a severe traumatic event, above all collisions with a vehicle, while many of those in Group A were hospitalised with minor or no injuries. Data on the main causes of roe deer hospitalisation are in line with previous studies [7,13,17,34,35,36,37,38].

The results of the haematological and biochemical parameters for the weaned groups were compared with the literature [19,28,29,30].

Both groups (deceased and survivors) showed lower HCT values (39.7% and 43.4% vs. 52% [30], higher MCV (42.8 fL and 43.9 fL vs. 35–40 fL [19,28,30]), CK (9720 IU and 4410 IU vs. 420–3559 IU [19,28,29,30]), and urea (10.1 mmol/L and 12.2 mmol/L vs. 2.6–7.8 mmol/L [19,28,29,30]). Although within the reference range, the values obtained in both groups were also close to the upper reference limit for creatinine (114.9 umol/L and 159.2 umol/L vs. 53.5–176.8 umol/L [19,28,29,30]). The other haematological parameters, including those relating to the leukocyte differential count and biochemical parameters, appear to have values in line with those reported in the literature [19,28,30].

The animals enrolled in the previous studies were all healthy, while our animals were all traumatised; thus, the different animals included (healthy vs. traumatised) may explain the differences found.

The low HCT could be due to haemorrhages related to the traumatic lesions that affected most of the animals in Group B. The high MCV values could be due to the release of young macrocytic erythrocytes or even reticulocytes from the spleen and bone marrow that occurs during regenerative anaemia following severe blood loss or related to the electrolyte imbalance of injured animals [39,40,41,42,43,44,45].

The increase in CK could be related to the capture stress response and the muscle damage induced both by trauma and handling during rescue and transport. In wild ungulates, stress and forced restraint can lead to capture myopathy, which is characterised by myositis, rhabdomyolysis, and a consequent increase in the related enzymes, such as CK, which is the most sensitive and specific marker [18,20,46].

The increase in urea may also be due to prolonged exertion and stress resulting from capture due to the physical exercise, the diminished renal perfusion, and the effect of glucocorticoids over protein catabolism [28,47].

With regard to the unweaned group (Group A), a similar comparison was not possible, as in the literature, there are no specific reference intervals for roe deer at this age.

Overall, we found statistical differences in the haematological and biochemical parameters between the deceased and survived animals, both for the unweaned and weaned groups. For the unweaned group, a higher number of both statistically different haematological and biochemical parameters were found between the deceased and survived groups, while only total bilirubin in the weaned animals differed between the two groups, possibly due to the different causes of hospitalisation. In fact, the survived unweaned were either slightly injured or not injured, whereas the deceased unweaned were severely traumatised. The clinical conditions upon arrival at the hospital differed considerably. On the other hand, upon admission, in the weaned groups, most of both the survived and the deceased animals had been involved in a traumatic event with similar compromised clinical conditions.

Statistical differences were found between Group A1 and A2 for MCV, MCH, CK activities, creatinine, total, direct and indirect bilirubin concentrations, while differences were only detected for total bilirubin between Group B1 and B2.

The MCV and MCH values were higher in Group A1 than in group A2, and with significant differences. This increase in MCV could be related to an electrolyte imbalance (mainly Na and K, including dehydration) in Group A1 (i.e., deceased animals).

On the other hand, the MCH values may have been high due to haemolysis or myoglobin derived from the muscle injury, as highlighted by the statistically significant increase in CK found in this study or to the increase in total bilirubin [33]. Again, in the latter case, there was a statistically significant higher increase in Group A1 than in Group A2. There was no haemolysis in samples from Group A1 animals and no correlation was found between the MCH and CK values, either for Group A1 or A2. However, a correlation was found between MCH and total bilirubin in Group A1 and A2; thus, the increase in MCH appeared to be due to artefactual interference.

CK is a highly sensitive and specific serum indicator of striated muscle damage in domestic animals and an increase in CK activity is likely associated with rhabdomyolysis due to muscle damage (i.e., trauma or exhaustion). Our results showed higher CK activity in the A1 group than in the A2 group. Group A1 was mainly composed of traumatised animals in which the muscle damage was severe. The differences in CK, but not AST, activity between the two groups may be due to the small number of animals included or because CK usually increases within hours, while AST increases 24–48 h after muscle damage, and, therefore, AST is less sensitive to mild and moderate insult than CK [39,40,46,47]. The animals included were rescued within hours and not days; thus, the increased CK activity seems to have been related to acute muscle damage.

The creatinine concentration was higher in Group A1 than A2. The increase of creatinine may be related to various causes that can induce acute prerenal or renal failure. Firstly, fawns need several meals to maintain energy intake and hydration. The unweaned fawns in Group 1 were severely traumatised; thus, they might have been unable to feed from the mother during transport from the rescue area to the hospital. A lower intake of fluid causes dehydration and impaired renal blood flow with an increase in serum creatinine. Moreover, stress resulting from capture and handling during the rescue can cause a decrease in renal blood flow because of vasospasm in the kidney vessels due to catecholamines [20,46]. Lastly, trauma and stress-related muscle exhaustion are the main causes of rhabdomyolysis, which entails a massive release of myoglobin in the blood. This metabolite is excreted by the kidney, but is toxic for kidney tubules if the animal is also dehydrated [20,46]. In our study, the creatinine, but not urea concentration, was statistically higher in Group 1 than Group 2; however, in ruminants, a higher creatinine concentration is a more reliable indicator of renal failure than urea concentration. 

The main causes of a higher bilirubin concentration in traumatised patients are the breakdown of erythrocyte, which releases haemoglobin during haemolysis or intracavitary haemorrhage, and hepatic failure caused by sepsis, infection, shock, and systemic hypotension [48,49,50,51,52]. Patients in both Groups A1 and B1 were severely traumatised; thus, the most plausible cause of the hyperbilirubinemia may have been the degradation of the blood components in the injured soft tissues, and bone fracture sites and body cavities. Moreover, stress from restraint and transport can increase the bilirubin concentration, as already reported in calves and wild ruminants [53,54].

No statistical differences between Group B1 and B2 were observed, except for the higher total bilirubin in B1 than in B2. This is likely due to the low number of animals in Group B2 and because most of the unweaned animals were affected by severe traumatic lesions in both Group B1 and B2, with consequently similar critical clinical presentation.

This study has some limitations. First, there are no reference values available that are subtyped for age in roe deer. In addition, since we are a veterinary hospital and can only study sick patients, we were unable to include a control population of healthy animals to make comparisons. In any case, we compared our results with findings reported in the literature on healthy animals.

Moreover, given that wild animals may have to undergo sedation or anaesthesia in order to restrain them, this alters the blood parameters, such as the erythrocyte and leukocyte values, due to a decrease in blood pressure and spleen involvement to shift these cells.

Lastly, the results may have been affected by bias because a few animals enrolled in the deceased group (both unweaned and weaned) were euthanised not because of their clinical condition, but because they had lesions that were not compatible with a positive release back into the wild.

## 5. Conclusions

The main result of our study is that the survival rate in our unweaned group was higher than in the weaned groups because the unweaned group presented fewer severe injuries.

Another important finding is that CK, creatinine, and bilirubin are parameters that should be considered in traumatised animals. Indeed, the lower values identified in both groups of surviving roe deer seem to be useful in predicting the severity of trauma and clinical condition. Further studies are needed to verify this use including a higher number of animals and using more homogeneous groups.

Blood work should be performed in all roe deer that are hospitalised to obtain a grading of the severity of the injuries received, which would thus help in managing the treatment and predicting the prognosis.

## Figures and Tables

**Table 1 animals-12-03469-t001:** Subtyping into groups according to roe deer age (unweaned or weaned) and hospitalisation outcome (deceased or survived).

	Deceased	Survived	Total
**Unweaned**	*Group A1* 9/16 (56%)	*Group A2* 7/16 (44%)	16/98 (16.3%)
**Weaned**	*Group B1* 68/82 (68%)	*Group B2* 14/82 (32%)	82/98 (83.7%)
**Total**	77/98 (78%)	21/98 (22%)	98

**Table 2 animals-12-03469-t002:** Reason for hospitalisation in the cohort of 98 patients and categorised into groups. Group A1: deceased unweaned; Group A2: survived unweaned; Group B1: deceased weaned; Group B2: survived weaned.

	Group A1	Group A2	Group B1	Group B2	Total
**Collision with vehicle**	2/9 (22%)	1/7 (14%)	57/68 (84%)	10/14 (71%)	70/98 (71%)
**Entrapment in nets**	1/9 (11%)	-	2/68 (3%)	1/14 (7%)	4/98 (4%)
**Trauma from agricultural machinery**	5/9 (56%)	-	1/68 (1%)	-	6/98 (6%)
**Predation**	-	-	1/68 (1%)	-	1/98 (1%)
**Unjustified rescue**	1/9 (11%)	3/7 (43%)	-	-	4/98 (4%)
**Systemic diseases**	-	-	1/68 (1%)	-	1/98 (1%)
**Unknown**	-	3/7 (43%)	6/68 (9%)	3/14 (22%)	12/98 (13%)
**Total**	9	7	68	14	98

**Table 3 animals-12-03469-t003:** Clinical lesions diagnosed in the cohort of 98 patients and categorised into groups. Group A1: deceased unweaned; Group A2: survived unweaned; Group B1: deceased weaned; Group B2: survived weaned.

	Group A1	Group A2	Group B1	Group B2	Total
**Limb fracture/luxation**	4/9 (45%)	1/7 (14%)	18/68 (27%)	1/14 (7%)	24/98 (21%)
**Rib fracture**	-	-	4/68 (6%)	-	4/98 (4%)
**Pelvis fracture**	-	-	6/68 (9%)	-	6/98 (6%)
**Head trauma**	-	1/7 (14%)	3/68 (4%)	4/14 (29%)	8/98 (8%)
**Abdominal trauma**			3/68 (4%)	1/14 (7%)	4/98 (4%)
**Vertebral fracture/luxation**	1/9 (11%)	-	10/68 (15%)	-	11/98 (11%)
**Soft tissue wounds (skin, muscle)**	2/9 (22%)	3/7 (43%)	3/68 (4%)	8/14 (57%)	16/98 (6%)
**Multiple traumas**	2/9 (22%)	-	21/68 (31%)	-	23/98 (24%)
**No lesions**	-	2/7 (29%)	-	-	2/98 (2%)
**Total**	9/98	7/98	68/98	14/98	98

**Table 4 animals-12-03469-t004:** Haematological parameters and differential cell counts assessed in the cohort of 86 patients categorised into groups.

	Group A1 *n* = 8		Group A2 *n* = 6		Group B1 *n* = 59		Group B2 *n* = 13	
**RBC (M/mL)**	8.6 4.1–12.8		7.5 6.7–9.8		10.1 0.9–14.1		9.4 8.2–10.5	
**HCT (%)**	45.4 18.1–62.6		31.6 24.3–41.2		43.4 3.9–63.1		39.7 32.9–45.9	
**HGB (g/dL)**	14 4.4–20.1		9.5 8.1–14.2		14.1 1.5–20.8		13.7 11.7–16	
**MCV (fL)**	45 * 31.0–55.4		41.5 ** 33.0–44.9		42.8 32.6–50		43.9 39.8–47.3	
**MCH (pg)**	14.6 * 7.5–17.1		12.7 ** 11.1–14.5		14.2 10.7–16.3		14.5 13.7–15.6	
**MCHC (g/dL)**	32.1 24.3–37.2		31.1 29.2–35.3		33.2 26.8–38.5		34.5 31.9–35.6	
**PLT (K/mL)**	301 102–832		506 231–664		182 7–514		194.5 74–203	
**WBC (K/mL)**	7.5 2.2–11.4		3.8 2.5–8.9		4.6 1.1–12.4		6.5 1.6–8	
	**10^9^/L**	**%**	**10^9^/L**	**%**	**10^9^/L**	**%**	**10^9^/L**	**%**
**Neutrophil**	5.1 0.9–8.2	69 26–76	1.9 0.6–6.9	47 24–78	2.4 0.2–9	55 8–96	2.5 0.6–6.6	53 28–98
**Band neutrophil**	0.1 0–0.6	1.5 0–15	0 0–0	0 0–0	0 0–1.6	0 0–16	0 0–2.6	0 0–24
**Lymphocyte**	1.5 0.7–2.5	25 10–54	1.7 1.2–2.7	50 13–59	1.6 0–4.3	32 2–82	2.2 0–4.1	36 0–62
**Monocyte**	0.2 0.1–0.7	5 1–8	0.5 1–0.8	9 3–25	0.1 0–1.3	4 0–23	0.2 0–0.7	4 0–10
**Eosinophil**	0 0–0	0 0–0	0 0–0	0 0–0	0 0–2.9	0 0–54	0.1 0–0.5	2 0–10
**Basophil**	0 0–0	0 0–0	0 0–0	0 0–0	0 0–0.3	0 0–4	0 0–0	0 0–1

Notes: Group A1: deceased unweaned; Group A2: survived unweaned; Group B1: deceased weaned; Group B2: survived weaned. All values are reported as median, minimum, and maximum. * ≠ ** Statistical differences between A1 vs. A2 group for MCV (*p* = 0.0350) and MCH (*p* = 0.0029). Legend—RBC: red blood cell count; Hct: haematocrit; Hgb: haemoglobin, MCV: mean corpuscular volume; MCH: mean corpuscular haemoglobin; MCHC: mean corpuscular haemoglobin concentration; WBC: white blood cell count; PLT: platelet count; %: percentage.

**Table 5 animals-12-03469-t005:** Biochemical parameters assessed in the cohort of 98 patients categorised into groups.

	Group A1 *n* = 9	Group A2 *n* = 7	Group B1 *n* = 68	Group B2 *n* = 14
**GGT (U/L)**	92.6 57.4−563.6	104 50–479	92.3 30–593.5	78 52.9–185
**AST (U/L)**	262 2–8865	54 33–18,000	964 133–14,272	397 282–1578
**CK (U/L)**	8865 * 116–19,538	108 * 70–501	9720 106–166,960	4410 1617–8148
**UREA (mmol/L)**	9.8 1.3–34.1	10.1 0.3–25	12.2 0.7–76.1	10.1 4.3–21
**CREA (umol/L)**	123.8 * 61.9–353.7	79.6 ** 70.7–114.9	159.2 88.4–486.3	114.9 79.6–265.3
**TOT BIL (umol/L)**	18.8 * 3.4–218.9	3.4 ** 3.4–6.8	15.4 ^§^ 1.7–218.9	11.9 ^§§^ 5.1–30.8
**DIR BIL (umol/L)**	5.1 * 1.7–102.6	1.7 ** 1.7–11.9	3.4 0–116.3	1.7 1.3–30.8
**IND BIL (umol/L)**	3.4 * 1.5–116.3	1.7 ** 1.7–130	10.2 0–116.3	8.5 5.1–10.2
**TP (g/L)**	59 49–74	55 50–70	64 42–101	71 53–97

Notes: Group A1: deceased unweaned; Group A2: survived unweaned; Group B1: deceased weaned; Group B2: survived weaned. All values are reported as median, minimum, and maximum. * ≠ ** Statistical differences between A1 vs. A2 group for CK (*p* = 0.0006), creatinine (*p* = 0.0182), total, direct, and indirect bilirubin (*p* = 0.0025; *p* = 0.0120; *p* = 0.0123, respectively). ^§^ ≠ ^§§^ Statistical differences between B1 vs. B2 group for total bilirubin (*p* = 0.0206). Legend—GGT: gamma glutamyl-transferase; AST: aspartate aminotransferase; CK: creatine kinase; CREA: creatinine; TOT BIL: total bilirubin; DIR BIL: direct bilirubin; IND BIL: indirect bilirubin; TP: total protein.

## Data Availability

The data presented in this study are available on request from the corresponding author.

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
