# Peer review of "Relationship between Blood Parameters and Outcome in Rescued Roe Deer"

_animals, 2022, doi:10.3390/ani12243469_

Round 1
Reviewer 1 Report
The topic of the manuscript is potentially useful for the readers and some of the results obtained are interesting but it suffers of some important flaws mainly due to the lack of some important data (leukocyte differential counts) and overall to the overinterpretation of the results. The most important issue is related to the low number of cases for each group and the extreme heterogeneity of groups. The authors speculated about some possible relationship between results of bloodworks and outcome but these are fatally biased by the different composition of groups in terms of causes of admission (for instance the majority of fawns in A1 group are found after polithraumatism due to agricolture machines while in group A2 mainly the causes of recovery are much softer). I agree that bloodwork may be useful to frame the health status of recovered roe deer but the authors should avoid to overinterpret their results in terms of possible use of bloodworks to predict prognosis, at least until the groups are composed in a homogeneous matter. Also some of the possible interpretations given to the differences found among groups are speculative and not sounding from a clinical pathologic point of view.
Some additional references should be also considered for discussion and comparison of data:
1) Zele and Vengust, Biochemical indicators in serum of free-ranging roe deer
(Capreolus capreolus) in Slovenia, ACTA VET. BRNO 2012, 81: 377–381
2) Montanè et al., EFFECTS OF ACEPROMAZINE ON CAPTURE STRESS IN ROE
DEER (CAPREOLUS CAPREOLUS), Journal of Wildlife Diseases, 39(2), 2003, pp. 375–386
3) Di Lorenzo et al., Blood L-Lactate concentration as an indicator of outcome in roe deed admitted to a wildlife rescue center, Animals, 2020, 10(6) 1066
Specific issues:
Lines 93-100: the splitting of results in different survival groups is not clear. To this reviewer's experience in a rescue center a huge number of animals are generally euthanized because they are not compatible with a positive release in nature and not always due to their possibility to live, once cured. This is for instance the case of deer which should be submitted to amputation which are not compatible with release and which are generally euthanized. What about this kind of animals? Were they included in the non surviving group? And what about the same kind of animals which were amputated and given in custody to authorized people since they are not compatible with a full release in wild? This point may be a source of bias among groups.
Lines 111-113: were all blood samples withdrawn before any therapy including fluids? This is an important point for the further interpretation of results, including in particular effects of dehydration on RBC variables.
Line 113: the issue of hemolysis is crucial for bloodworks in wild animals. Please specifiy how hemolysis was checked and evaluated and which samples were considered unsuitable and discarded (what does grossly hemolytic mean? How many samples were discarded due to hemolysis?)
Line 115-120: why the authors did not report results for leukocyte differential counts? Procyte provides them, even if is not validated for the use in roe deer. However, it is likely that it can be validated starting from another ruminant species setting (bovine or sheep) with a little effort. This should be an important piece of information to better frame the health status and interpret results. As a possible alternative, authors could have performed a manual differential count on a stained smear.
Line 191: the authors stated that their survival results confirmed those of a previous study of the same research group. Were the cases included in the present manuscript different from those included in the cited paper (ref 6)? Otherwise it is silly to state that the results comfirm themselves! This should be clarified.
Lines 201-203: The authors stated that increased in MCV could be due to the release of reticulocyte and cite two references (not listed in the references list) but I supposed obtained in other species than roe deer. Do they have any information about the possible release of reticulocytes in circulation in roe deer? Reticulocytes are generally not found in peripheral blood of species with a RBC half life longer than 120 days and this is the case of many ruminants species including some cervids. Do the authors have the evidence that reticulocytes may be found in roe deer blood in case of regeneration to anemia?
Lines 216-217: I agree that the differences found between groups A1 and A2 are related to the different composition of causes of recovery with A2 group composed just of slightly or not injured fawns. All the other possible comments about differences are speculative and not supported by the results of the present study.
Lines 235-237: the authors suggest that the higher values of MCH are likely related to artifactual interference of bilirubin, however concentrations of bilirubin in the present dataset are not so high to support a possible artifactual interference (at least as mean values). Do the authors have any information about the cut off values for bilirubin that should considered at risk of artifactual overestimation of hemoglobin (and all related derived variables such as MHC) using Procyte? Otherwise this speculation is not supported.
Line 244: please provide a reference
Line 262: what do the authors mean for "breakdown of blood components"? Hemolysis (not likely in the present caseload? Reabsorption of hemoglobin from hemorrages in internal cavities? Else? This should have been specified.
Author Response
Reviewer 1
The topic of the manuscript is potentially useful for the readers and some of the results obtained are interesting, but it suffers of some important flaws mainly due to the lack of some important data (leukocyte differential counts) and overall, to the overinterpretation of the results. The most important issue is related to the low number of cases for each group and the extreme heterogeneity of groups. The authors speculated about some possible relationship between results of bloodwork and outcome, but these are fatally biased by the different composition of groups in terms of causes of admission (for instance the majority of fawns in A1 group are found after polithraumatism due to agricolture machines while in group A2 mainly the causes of recovery are much softer). I agree that bloodwork may be useful to frame the health status of recovered roe deer, but the authors should avoid to overinterpret their results in terms of possible use of bloodwork to predict prognosis, at least until the groups are composed in a homogeneous matter. Also, some of the possible interpretations given to the differences found among groups are speculative and not sounding from a clinical pathologic point of view.
Some additional references should be also considered for discussion and comparison of data:
1) Zele and Vengust, Biochemical indicators in serum of free-ranging roe deer
(Capreolus capreolus) in Slovenia, ACTA VET. BRNO 2012, 81: 377–381
2) Montanè et al., EFFECTS OF ACEPROMAZINE ON CAPTURE STRESS IN ROE
DEER (CAPREOLUS CAPREOLUS), Journal of Wildlife Diseases, 39(2), 2003, pp. 375–386
3) Di Lorenzo et al., Blood L-Lactate concentration as an indicator of outcome in roe deed admitted to a wildlife rescue center, Animals, 2020, 10(6) 1066.
Answer: the papers suggested have been added and discussed.
Specific issues:
Lines 93-100: the splitting of results in different survival groups is not clear. To this reviewer's experience in a rescue center a huge number of animals are generally euthanized because they are not compatible with a positive release in nature and not always due to their possibility to live, once cured. This is for instance the case of deer which should be submitted to amputation which are not compatible with release, and which are generally euthanized. What about this kind of animals? Were they included in the non-surviving group? And what about the same kind of animals which were amputated and given in custody to authorized people since they are not compatible with a full release in wild? This point may be a source of bias among groups.
Answer: we agree. We have added in the deceased groups (both for fawns and yearlings/adults) the roe deer that died spontaneously or were euthanized because they were close to death or because the lesions were not compatible with a positive release in the wild. The sentence has rewritten (please see lines 99-100). We had this type of animals in both groups (fawns and yearlings/adults). We have thus added the possible bias to the study limitations (please see lines 303-305).
Lines 111-113: were all blood samples withdrawn before any therapy including fluids? This is an important point for the further interpretation of results, including in particular effects of dehydration on RBC variables.
Answer: blood samples were always collected before any therapy. This has been added to the main text (please see line 115).
Line 113: the issue of hemolysis is crucial for bloodwork in wild animals. Please specify how hemolysis was checked and evaluated, and which samples were considered unsuitable and discarded (what does grossly hemolytic mean? How many samples were discarded due to hemolysis?)
Answer: the definition “grossly hemolyzed” has been delated and the sentence has been reworded and a reference has been added (please see lines 119-121). The excluded number of samples has been added to the results (please see lines 172).
Line 115-120: why the authors did not report results for leukocyte differential counts? Procyte provides them, even if is not validated for the use in roe deer. However, it is likely that it can be validated starting from another ruminant species setting (bovine or sheep) with a little effort. This should be an important piece of information to better frame the health status and interpret results. As a possible alternative, authors could have performed a manual differential count on a stained smear.
Answer: Pro-cyte shows the WBC differential count for all samples, but the values have not been validated by Idexx or specific research on roe deer. We also performed a manual differential count on stained smear, but the interpretation of these results is difficult as there are few data on the reference ranges and we do not know the leukogram pattern in the disease so the conclusion can only speculative.
Line 191: the authors stated that their survival results confirmed those of a previous study of the same research group. Were the cases included in the present manuscript different from those included in the cited paper (ref 6)? Otherwise, it is silly to state that the results confirm themselves! This should be clarified.
Answer: the sentence has been changed (please see line 205).
Lines 201-203: The authors stated that increased in MCV could be due to the release of reticulocyte and cite two references (not listed in the references list), but I supposed obtained in other species than roe deer. Do they have any information about the possible release of reticulocytes in circulation in roe deer?
Answer: In fact, little is known in roe deer about the reticulocyte behavior. We speculated that the MCV could be related to a severe hemorrhagic anemia which stimulates the new and young erythrocytes (generally macrocytic as in Cervidae) stored in the spleen and bone marrow to enter in the peripheral bloodstream. However, the presence of reticulocytes in the circulation is documented in ruminants and also in cervids and we have added the specific literature reference in the text (please see line 221). We have also added that the MCV could be influenced by the electrolyte imbalance.
Reticulocytes are generally not found in peripheral blood of species with a RBC half-life longer than 120 days and this is the case of many ruminants species including some cervids. Do the authors have the evidence that reticulocytes may be found in roe deer blood in case of regeneration to anemia?
Answer: see previous observation.
Lines 216-217: We completely agree regarding the differences related to the different composition of those two groups. In any case, we believe that a possible explanation for the altered parameters is important. In our opinion, the hypothesis reported is plausible since it is related to the severity of trauma that affected the deceased and the survived.
Answer: we completely agree with the reviewer regarding differences related to different composition of those two groups. Anyway, we think that a possible hypothesis about the reason why the parameters may be altered is mandatory. In our opinion, the hypothesis reported are anyway plausible since related with severity of trauma that affected the deceased and the survived.
Lines 235-237: the authors suggest that the higher values of MCH are likely related to artifactual interference of bilirubin, however concentrations of bilirubin in the present dataset are not so high to support a possible artifactual interference (at least as mean values). Do the authors have any information about the cut off values for bilirubin that should considered at risk of artifactual overestimation of hemoglobin (and all related derived variables such as MHC) using Procyte? Otherwise, this speculation is not supported.
Answer: As occurs in other species, the MCH is related to the assessment of hemoglobin inside the erythrocyte + the interference due to hemoglobin and bilirubin in plasma or from some other artifacts in erythrocytes (i.e. Heinz bodies, eccentrocytes, spherocytes) (see the CLSI reference, line 250).
Line 244: please provide a reference.
Answer: provided (see line 263).
Line 262: what do the authors mean for "breakdown of blood components"? Hemolysis (not likely in the present caseload? Reabsorption of hemoglobin from hemorrhages in internal cavities? Else? This should have been specified.
Answer: we have corrected the sentence (please see line 281).

Reviewer 2 Report
This is an interesting paper and could add to the relatively limited amount of information available to those treating injured roe deer. As the authors suggest, such information is useful to veterinarians dealing with clinical cases. The paper is generally well presented, and the use of English is extremely good. There has been a reasonably good review of the available literature, but more could be made of directly comparing the available published material with the findings of the authors. More could also be made of the benefit of rapid assessment in wild animals in order to enable rapid decision-making, especially relating to treatment or euthanasia, and ensure animal welfare. Some specific comments are given below.
Line 2 Does the title properly capture what this paper is about? Some of these animals were not hospitalized (they were rightly quickly euthanased) and samples were taken at admission, before treatment and hospitalization was undertaken.
Line 10 This is several tests not just one, also not all tests are quick. Maybe change to “Blood tests can be a quick and minimally invasive way of gathering useful clinical information” or similar.
Line 14 Not sure ‘extensive’ is correct, what was carried out was fairly routine. I’d remove ‘extensive’
Line 15 Should this be ‘within’ both groups rather than ‘between’? Between suggests in the fawn group compared to the yearling/adult group.
Line 16 Are you confident that all three biochemical markers are useful? Even in the abstract I might try and clarify which were significantly different.
Maybe change to ‘Creatine 15 kinase, creatinine and bilirubin may be useful indicators that correlate with severity of trauma and help predict prognosis.’ (see also line 31)
Line 20 Why is all this important? Welfare perhaps?
Line 21/22 As line 10
Line 23 I might say ‘…., and to help predict hospitalization outcomes.’ I’d hope they were taken in conjunction with other factors, such as a good clinical examination.
Line 26 ‘…, to discuss their values…..’ is probably unnecessary in this sentence
Line 31 Maybe change to something like ‘CK, creatinine and bilirubin may be useful indicators that correlate with severity of trauma and help predict prognosis.’
Line 48 Could the references be allocated to specific points you are trying to make? Same on line 55 and 62
Line 51 Are the ‘health problems’ all trauma related? The term ‘health problems’ perhaps makes these sound like diseases, which I suspect is not what you mean.
Line 55 ‘fawn’ not cub
Line 62 Are there not records after 2008? These are quite old figures to illustrate a current trend.
Line 71 Again references could be allocated better to specific points you are making.
Line 74 Is there a reference for the increasing trend in admissions?
Lines 79-83 Consider reworking, to make it clear that blood tests are additional clinical information used in determining likely outcome. For example: “In conjunction with examination, haematological and biochemical blood parameters can provide additional clinical information which can be used to determine case specific treatment, management and prognosis following hospitalization.” Wildlife rehabilitators usually refer to this decision-making process as ‘triage’. You could also say why this is important, especially in wildlife casualties where the stress of captivity can be a welfare concern.
Line 94 Is a 5mth old animal a ‘yearling’ or an ‘adult’? Maybe your division of animals is just ‘weaned’ and unweaned’?
Line 96 Were only ‘near death’ animals euthanased? What about ones that just had no reasonable chance of recovery and eventual release?
Line 107 In the discussion I think you need to fully consider the possible impact of handling and of any chemical sedation. I’m not suggesting that this was wrong, but you need to acknowledge the possible impact. You blood results could be affected by these things as well as by the main reason for admission.
Line 116 How useful and accurate is the Procyte for running ‘other species’ haematology? Are you confident it is reliable? In house analysers are often not great for haematology on unusual species that they are not set up for. Maybe check with Idexx what any errors are likely to be. Biochemistry is usually always OK on in house analysers.
Lines 124-130 Why were these parameters chosen? Was it just a standard biochemistry profile? Why not other parameters, lactate in particular.
Was there any comparison of the values obtained with known reference ranges for roe deer?
Table 2
Column 1 (Group A1) The trauma from agricultural machinery needs to be 56% in order to make the whole column 100%
Column 3 (Group B1) Does group B1 not have 56 animals in it, as in Table 1? Where is the 68 figure from?
Column 4 (Group B2) Does group B2 not have 26 animals in it, as in Table 1? Where is the 14 figure from?
Column 5 (Total) 70/98 is 71% as in your footnote
Table 3
Same comments as for Table 2 around the total number of animals in groups B1 and B2
Please check the percentage calculations
Lines 157/158 The percent figures here would be better as a reflection of the percent of animals in each group samples e.g. 8/16 (50%) in group A1
Line 158 Need a percent figure for 12/98
Tables 4 and 5 Could these tables be better laid out with the ‘Group’ headings on the LHS or even at the top (as in tables 2 and 3) with the parameters down the side? It might then be possible to make the ‘mean’ then ‘minimum-maximum’ values clear.
Line 194 Relating the historical references to some of your specific findings would make this more interesting and valid.
Line 196 This suggests that you compared your findings to reference ranges. Which reference ranges did you use? Would it not be useful to include these comparisons in your results?
Line 204 Comparison of the CK levels you found with those reported in the literature would be of interest here.
Line 228 If animals were dehydrated would the HCT not have been increased?
Line 273 These ‘limitations’ perhaps need to be put into some context, how many animals were chemically restrained for example?
Lines 278-280 This should probably be in materials and methods
This paper may be of interest and usefulness
Di Lorenzo, E.; Rossi, R.; Ferrari, F.; Martini, V.; Comazzi, S. Blood L-Lactate Concentration as an Indicator of Outcome in Roe Deer (Capreolus capreolus) Admitted to a Wildlife Rescue Center. Animals 2020, 10, 1066. https://doi.org/10.3390/ani10061066
Author Response
Reviewer 2
This is an interesting paper and could add to the relatively limited amount of information available to those treating injured roe deer. As the authors suggest, such information is useful to veterinarians dealing with clinical cases. The paper is generally well presented, and the use of English is extremely good. There has been a reasonably good review of the available literature, but more could be made of directly comparing the available published material with the findings of the authors. More could also be made of the benefit of rapid assessment in wild animals in order to enable rapid decision-making, especially relating to treatment or euthanasia, and ensure animal welfare. Some specific comments are given below.
Line 2. Does the title properly capture what this paper is about? Some of these animals were not hospitalized (they were rightly quickly euthanized) and samples were taken at admission, before treatment and hospitalization was undertaken.
Answer: we consider all the roe deer as hospitalized because they have been admitted at the hospital, recorded on our database, a diagnostic imaging is usually performed (ultrasound, rx) and blood work processed. However, we have changed hospitalized to rescued.
Line 10. This is several tests not just one, also not all tests are quick. Maybe change to “Blood tests can be a quick and minimally invasive way of gathering useful clinical information” or similar.
Answer: thank you for the suggestion. The sentence has been reworded (please see line 10).
Line 14. Not sure ‘extensive’ is correct, what was carried out was fairly routine. I’d remove ‘extensive’.
Answer: done.
Line 15. Should this be ‘within’ both groups rather than ‘between’? Between suggests in the fawn group compared to the yearling/adult group.
Answer: done (please see line 15).
Line 16. Are you confident that all three biochemical markers are useful? Even in the abstract I might try and clarify which were significantly different. Maybe change to ‘Creatine 15 kinase, creatinine and bilirubin may be useful indicators that correlate with severity of trauma and help predict prognosis.’ (See also line 31).
Answer: done (please see lines 15-17 and 31-32).
Line 20. Why is all this important? Welfare perhaps?
Answer: yes, in our opinion it is mandatory to improve the clinical management of the wildlife rescue in order to improve the animal welfare. We have added a sentence to better explain this (please see line 19).
Line 21/22. As line 10.
Line 23. I might say ‘…., and to help predict hospitalization outcomes.’ I’d hope they were taken in conjunction with other factors, such as a good clinical examination.
Answer: the sentences have been rephrased (please see lines 21-24).
Line 26. ‘…, to discuss their values…..’ is probably unnecessary in this sentence.
Answer: done.
Line 31. Maybe change to something like ‘CK, creatinine and bilirubin may be useful indicators that correlate with severity of trauma and help predict prognosis.’
Answer: done.
Line 48. Could the references be allocated to specific points you are trying to make? Same on line 55 and 62.
Answer: done in line 62. Regarding lines 48 and 55, it is difficult to decide where exactly to place the references since the sentence is a fusion of the information taken from all the cited studies.
Line 51. Are the ‘health problems’ all trauma related? The term ‘health problems’ perhaps makes these sound-like diseases, which I suspect is not what you mean.
Answer: the sentence has been clarified (please see line 51).
Line 55. ‘fawn’ not cub.
Answer: corrected.
Line 62. Are there not records after 2008? These are quite old figures to illustrate a current trend.
Answer: specifically for Tuscany, there are records only for this period, but the trend regarding car accidents in Europe is increasing. We have added some literature about Europe and have specified that the records regarding Tuscany are old, but no more recent data are available.
Line 71. Again, references could be allocated better to specific points you are making.
Answer: done.
Line 74. Is there a reference for the increasing trend in admissions?
Answer: done.
Lines 79-83. Consider reworking, to make it clear that blood tests are additional clinical information used in determining likely outcome. For example: “In conjunction with examination, hematological and biochemical blood parameters can provide additional clinical information which can be used to determine case specific treatment, management and prognosis following hospitalization.” Wildlife rehabilitators usually refer to this decision-making process as ‘triage’. You could also say why this is important, especially in wildlife casualties where the stress of captivity can be a welfare concern.
Answer: we have reworded the sentence as suggested (please see lines 81-84). In general, triage is used if many patients come into the ICU unit and the priority of treatment is needed based on the degree of urgency. In our case, just one animal/each time is hospitalized, thus we proceed with a clinical exam and so on without triage just because no other patients are admitted at the same time. As you report, sometimes the word “triage” is used as general decision-making process, but in our opinion, this is the not proper use of the word.
Line 94. Is a 5mth old animal a ‘yearling’ or an ‘adult’? Maybe your division of animals is just ‘weaned’ and unweaned’?
Answer: we agree and have changed the term throughout the main text.
Line 96. Were only ‘near death’ animals euthanized? What about ones that just had no reasonable chance of recovery and eventual release?
Answer: the sentence has been better explained (please see lines 98-99).
Line 107. In the discussion I think you need to fully consider the possible impact of handling and of any chemical sedation. I’m not suggesting that this was wrong, but you need to acknowledge the possible impact. You blood results could be affected by these things as well as by the main reason for admission.
Answer: we agree and have added this as a limitation of the study. Unfortunately, handling and pharmacological restrain are not all avoidable due to the need to examine the animals (restrain) and to act in compliance with animal welfare.
Line 116. How useful and accurate is the Procyte for running ‘other species’ hematology? Are you confident it is reliable? In house analyzers are often not great for hematology on unusual species that they are not set up for. Maybe check with Idexx what any errors are likely to be. Biochemistry is usually always OK on in-house analyzers.
Answer: We agree that the current in house, blood cell counter has not been validated for the specific species here reported. Generally, the blood count is acceptable, while the morphological evaluation is not reliable (i.e. WBC differential or reticulocyte count). We consulted technical support atIdexx, but so far we have not received any advice for this species.
Lines 124-130 Why were these parameters chosen? Was it just a standard biochemistry profile? Why not other parameters, lactate in particular. Was there any comparison of the values obtained with known reference ranges for roe deer?
Answer: this is our standard biochemistry profile. We have also begun to perform lactate using a blood-gas analyzer; however, we do not have the results on all this cohort. We thus decided not to add the parameter to this paper and to continue to collect data in order to write another paper with this analyte in the near future. We have compared our data with the reference range for roe deer (please see line 208-216).
Table 2
Column 1 (Group A1). The trauma from agricultural machinery needs to be 56% in order to make the whole column 100%.
Column 3 (Group B1) Does group B1 not have 56 animals in it, as in Table 1? Where is the 68 figure from?
Column 4 (Group B2) Does group B2 not have 26 animals in it, as in Table 1? Where is the 14 figure from?
Column 5 (Total) 70/98 is 71% as in your footnote.
Answer: the numbers and percentages have now been corrected (please see tables).
Table 3
Same comments as for Table 2 around the total number of animals in groups B1 and B2.
Please check the percentage calculations.
Answer: done (please see tables).
157/158. The percent figures here would be better as a reflection of the percent of animals in each group samples e.g. 8/16 (50%) in group A1.
Answer: corrected (please see lines 166-167).
Line 158. Need a percent figure for 12/98.
Answer: added (please see line 168).
Tables 4 and 5. Could these tables be better laid out with the ‘Group’ headings on the LHS or even at the top (as in tables 2 and 3) with the parameters down the side? It might then be possible to make the ‘mean’ then ‘minimum-maximum’ values clear.
Answer: done (please see tables).
Line 194. Relating the historical references to some of your specific findings would make this more interesting and valid.
Answer: done (please see lines xxx).
Line 196. This suggests that you compared your findings to reference ranges. Which reference ranges did you use? Would it not be useful to include these comparisons in your results?
Answer: we have already compared our results with the literature, and thus have rewritten the sentence to better explain the comparison (please see lines 208-216).
Line 204. Comparison of the CK levels you found with those reported in the literature would be of interest here.
Answer: done.
Line 228. If animals were dehydrated would the HCT not have been increased?
Answer: we agree, we found no differences between A1 vs. A2, however in both groups, the HCT was higher than the reference interval reported for healthy roe deer (please see lines 208-216).
Line 273. These ‘limitations’ perhaps need to be put into some context; how many animals were chemically restrained for example?
Answer: all animals were under sedation/general anaesthesia. The sentence has been rephrased (please, see line 109).
Lines 278-280 This should probably be in materials and methods.
Answer: the sentence has been reworded and shifted to the material and methods section (please see lines 115-117).
This paper may be of interest and usefulness:
Di Lorenzo, E.; Rossi, R.; Ferrari, F.; Martini, V.; Comazzi, S. Blood L-Lactate Concentration as an Indicator of Outcome in Roe Deer (Capreolus capreolus) Admitted to a Wildlife Rescue Center. Animals 2020, 10, 1066. https://doi.org/10.3390/ani10061066
Answer: thank you, the paper has been added to the references.

Reviewer 3 Report
Manuscript brings interesting data about health and physiological parameters of roe deer in Tuscany (Italy). After studying the material, I have some notes and requirements:
- According to what census was estimated the statement that Tuscany has 400.000 ungulates?
- How was estimated the limit for dividing the animals on two age classes? The second age group includes the individuals from 4 months of age to more years, the group is according to age very variable. It seems to be more suitable to divide the animals on two groups up to 1 year and older.
- Which method was used for age determination
- Animals with age near the limit value were included into evaluation.
- The main lack of work is the absence of results from the blood parameters of healthy roe deer living in free range - reference values
- Results are well described, supported with proper statistical method.
- The results are written very briefly authors should declare better the significance of the results for science and practice.
- As I am not the native speaker, I do not evaluate the quality of English.
- After some modification, I recommend the work for publication
Author Response
Reviewer 3
Manuscript brings interesting data about health and physiological parameters of roe deer in Tuscany (Italy). After studying the material, I have some notes and requirements:
According to what census was estimated the statement that Tuscany has 400.000 ungulates?
Answer: the number of ungulates in Tuscany, and throughout Italy, was estimated on the basis of the census carried out by ISPRA, the Italian Institute for Environmental Protection and Research.
How was estimated the limit for dividing the animals on two age classes? The second age group includes the individuals from 4 months of age to more years, the group is according to age very variable. It seems to be more suitable to divide the animals on two groups up to 1 year and older.
Answer: we divided the animals in this way in order to standardize the populations as much as possible based on the traumas suffered and therefore the causes of hospitalization. Before and after weaning, the animals have different behaviors, make different movements which predisposes them to different risks. In fact, in this study, it is pretty clear that there were different reasons for hospitalization between groups A and B.
Which method was used for age determination.
Answer: The age was determined based on morphological aspects and on the dental examination to determine the state of eruption and wearing down of the teeth. A sentence has now been added in the text (please see lines 95-97).
Animals with age near the limit value were included into evaluation.
Answer: yes, they were included.
The main lack of work is the absence of results from the blood parameters of healthy roe deer living in free range - reference values.
Answer: we agree. We are a veterinary teaching hospital, thus we can only study sick patients. We have compared our results with reference ranges found in the literature and have reported this limitation in the discussion (please see lines 293-295).
Results are well described, supported with proper statistical method.
Answer: Thank you very much.
The results are written very briefly authors should declare better the significance of the results for science and practice.
Answer: the significance of the results have been better explained in the discussion section.
As I am not the native speaker, I do not evaluate the quality of English.
Answer: the paper has been edited by a native English speaker and a declaration has been uploaded.
After some modification, I recommend the work for publication.

Round 2
Reviewer 1 Report
The authors adequately responded to many of my request and amended most major points. However, some issues still persist unsolved. The major point is that the authors tend to overestimate the potential prognostic use of their result for predicting outcome. This is stated both in abstract at line 30-31 and in the conclusion (lines 308-311). As already discussed in the previous round of reviewing, higher CK and/or bilirubin concentration in the group of surviving deer are related to the different causes and entity of thraumatic events and we cannot desume that they are markers for predicting prognosis unless the groups are composed by the same kind of thraumatic events. This cannot be derived from the data obtained in the present study. I suggest to remove any reference to the possible prognostic use of those analytes.
In addition, this reviewer is not satisfied about the response given by the authors regarding leukocyte differential counts: the authors stated that differential leukocyte counts were performed but not reported and discussed since “ the interpretation of these results is difficult as there are few data on the reference ranges and we do not know the leukogram pattern in the disease so the conclusion can only speculative”.
However, this statement may be true for all the analytes reported, since studies on hemato-biochemical results are limited in roe deer but it did not prevent from reporting results of other analytes and in some case attempting a discussion. Providing results of differential leukocyte counts may be useful for framing the overall pathological condition (inflammation, severe stress, endotoxic shock, etc) and to provide data for future comparison. If leukocyte differential were performed I suggest to report the results in table and to discuss them in the appropriate section (if any interesting emerges). Otherwise the authors should justify why this data are not available and address this as a limit of their study.
Some other minor issues are related to table 2 (one statement in the first column is lacking: “road collision” or something similar?) and table 4 (values of HCT for group B1 is not consistent with line 206).
Author Response
Reviewer: The authors adequately responded to many of my request and amended most major points. However, some issues still persist unsolved. The major point is that the authors tend to overestimate the potential prognostic use of their result for predicting outcome. This is stated both in abstract at line 30-31 and in the conclusion (lines 308-311). As already discussed in the previous round of reviewing, higher CK and/or bilirubin concentration in the group of surviving deer are related to the different causes and entity of thraumatic events and we cannot desume that they are markers for predicting prognosis unless the groups are composed by the same kind of thraumatic events. This cannot be derived from the data obtained in the present study. I suggest removing any reference to the possible prognostic use of those analytes.
Authors: the sentence has been rephrased, please see line 313-316.
Reviewer: In addition, this reviewer is not satisfied about the response given by the authors regarding leukocyte differential counts: the authors stated that differential leukocyte counts were performed but not reported and discussed since “the interpretation of these results is difficult as there are few data on the reference ranges and we do not know the leukogram pattern in the disease so the conclusion can only speculative”. However, this statement may be true for all the analytes reported, since studies on hemato-biochemical results are limited in roe deer but it did not prevent from reporting results of other analytes and in some case attempting a discussion. Providing results of differential leukocyte counts may be useful for framing the overall pathological condition (inflammation, severe stress, endotoxic shock, etc) and to provide data for future comparison. If leukocyte differential were performed, I suggest reporting the results in table and to discuss them in the appropriate section (if any interesting emerges). Otherwise, the authors should justify why this data are not available and address this as a limit of their study.
Authors: leukocyte differential counts have been added and discussed (please see table 4, line 126-130 and 214-216).
Reviewer: Some other minor issues are related to table 2 (one statement in the first column is lacking: “road collision” or something similar?) and table 4 (values of HCT for group B1 is not consistent with line 206).
Authors: the table and the sentence have been corrected (please see table 2 and line 209).
